# Assessment of cognitive screening tests as predictors of driving cessation: A prospective cohort study of a median 4-year follow-up

**Ioannis Kokkinakis**[1☯]*, **Paul Vaucher**[2,3☯], **Isabel Cardoso**[2,4], **Bernard Favrat**[1,2]

**1** Center for Primary Care and Public Health (Unisanté), University of Lausanne, Lausanne, Switzerland, **2** Traffic Medicine and Psychology Unit, University Center of Legal Medicine, Lausanne–Geneva, Lausanne University Hospital, Lausanne, Switzerland, **3** School of Health Sciences Fribourg, University of Applied Sciences and Arts Western Switzerland (HES-SO), Fribourg, Switzerland, **4** University Department of Advanced Age Psychiatry (SUPAA), Lausanne University Hospital, Lausanne, Switzerland

☯ These authors contributed equally to this work.
* ioannis.kokkinakis@unisante.ch

## Abstract

### Background

Assessing fitness to drive and predicting driving cessation remains a challenge for primary care physicians using standard screening procedures. The objective of this study was to prospectively evaluate the properties of neuropsychological screening tests, including the Trail Making Test (TMT), Clock Drawing Test (CDT), Montreal Cognitive Assessment (MoCA), Useful Field of View (UFOV), and Timed Up and Go (TUG) test, in predicting driving cessation for health reasons in drivers older than 70 years of age.

### Design and methods

This prospective cohort study, with a median follow-up of 4 years for drivers of 70 years old or older with an active driving license in Switzerland, included 441 participants from a driving refresher course dedicated to volunteer senior drivers. Cases were drivers reported in the national driving registry who lost their license following a health-related accident, who were reported as unfit to drive by their physician or voluntarily ceased driving for health reasons. Survival analysis was used to measure the hazard ratio of driving cessation by adjusting for age and sex and to evaluate the predictive value of combining 3 or more positive tests in predicting driving cessation during a 4-year follow-up.

### Results

A total of 1738 person-years were followed-up in the cohort, with 19 (4.3%) having ceased driving for health reasons. We found that participants with a TMT-A < 54 sec and TMT-B < 150 sec at baseline had a significantly lower cumulative hazard of driving cessation in 4 years than those with slower performance (adjusted HR 3, 95% CI: 1.16–7.78, p = 0.023). Participants who performed a CDT ≥ 5 had a significantly lower cumulative hazard of driving cessation (adjusted HR 2.89, 95% CI: 1.01–7.71, p = 0.033). Similarly, an MoCA score ≥

**Funding:** BF and PV were supported by a grant from the Department of Medicine and Community Health of the University Hospital of Lausanne, Switzerland. The funder had no role in study design, data collection and analysis, decision to publish, or preparation of the manuscript.

**Competing interests:** The authors have declared that no competing interests exist.

**Abbreviations:** TMT, Trail Making Test; UFOV, Useful Field of View; MoCA, Montreal Cognitive Assessment; CDT, Clock Drawing Test; MMSE, Mini Mental State Examination; TUG, Time Up and Go test; CI, Confidence Interval; MCI, Mild Cognitive Impairment.

26, TUG test <12 sec or a UFOV of low risk showed a lower but not significant cumulative risk at a median follow-up of 4 years. When using tests as a battery, those with three or more positive tests out of five were 3.46 times more likely to cease driving (95% CI: 1.31–9.13, p = 0.012).

## Conclusions

The CDT and the TMT may predict driving cessation in a statistically significant way, with a better performance than the UFOV and MoCA tests during a median 4-year follow-up. Combining tests may increase the predictability of driving cessation. Although our results are consistent with current evidence, they should be interpreted with precaution; more than 95% of the participants above the set threshold were able to continue driving for 4 years without any serious incident.

## Introduction

### Background/rationale

Fitness to drive in the geriatric population and the potential effects of cognitive and physical decline remain a difficult challenge for primary care physicians [1]. Research has shown that maintaining out-of-home mobility is of great importance for people moving to old age from late midlife [2]. Il is well established the association between driving cessation and functional dependency, depressive disorders, social dysfunction and mortality, with a considerable individual and societal impact [3]. As many adverse health problems have been related to driving cessation in later life, predicting and evaluating accurately the decline of driving capacity of older drivers is of critical importance [4]. Supporting and stimulating out-of-home mobility in the elder population, detecting and preventing a functional decline and possible future driving cessation, depends on individual screening strategies, as well as on transport policy and social policy measures [5].

Concerning the individual screening strategies, many clinical tests to evaluate the driving aptitude of patients, including cognitive, mental, motor and vision tests, have been proposed [1]. Among the promising neuropsychological tests, the **Trail Making Test (TMT-A and TMT-B)** and the **useful field of view (UFOV)** were estimated to be predictors of unsafe driving in a meta-analysis [6]. In older drivers, the TMT has a 63,6% sensitivity, 64,9% specificity, low positive predictive value (9,5%) and high negative predictive value (96,9%) for poor driving performance in a translational study [7]. Furthermore, TMT can be a useful tool in evaluating driving performance, as it can estimate various functions, such as visual scanning, executive function and graphomotor speed [8].

The **UFOV**, a computer-based test that evaluates visual processing speed and attention [9], has been developed and investigated as a tool to detect poor driving performance and has been correlated with increased risks of on-road accidents [10]. Normative values, based on measures taken from drivers referred to specialized centers for testing, have been used to define benchmarks for clinical practice [11]. It is, however, not clear how the instrument has managed to distinguish cognitive decline from visual reduction in contrast sensitivity or visual acuity, which also affects test and driving performance [12]. The literature also fails to clearly distinguish the old from the new version of the test, making it difficult to clearly assess its ability to predict on-road events [13].

Based on the current literature, cognitive tests alone have not shown a high predictive value in predicting unfit drivers and driving cessation [14]. Cognitive evaluation tests such as the **Montreal Cognitive Assessment (MoCA)** are considered to be useful screening tools to detect declining driving performance, but their predictive properties have shown low sensitivity (84.5%) and specificity (50%) with a cutoff score ≤25 [15]. Furthermore, a poor **clock drawing test (CDT)** performance has been correlated with a decline in driving aptitude evaluated by a driving simulation in a prospective cohort study [16]. A CDT score less than 5 out of 7 points was associated with significantly more driving errors and hazardous driving, indicating the need for a formal driving evaluation [16]. More recent studies show that the CDT could have limited screening predictive value as a solitary screening tool of driving performance [17]. The **Mini Mental State Examination (MMSE)** score has shown a small effect in predicting unsafe driving in older drivers [6]. Concerning the **Timed Up and Go (TUG) test,** which evaluates the ability to walk and balance safely and efficiently, the current literature shows that it could be a predictor of driving cessation (OR 12.60, CI 2.74–57.89; p < 0.01) [18]. A modified version of the TUG test has demonstrated high sensitivity for detecting fall risk in elderly individuals with good predictive values, rendering this test a possible predictor for elderly functional decline [19].

Although vision abilities, health indicators and physical capacities are usually considered to determine the driving aptitude, instrumental functional performance and cognitive speed of processing, indicators such as the UFOV were shown to be better predictors of driving cessation in a 3-year study [20]. In another 2-year cohort study, age and crash history were shown to indicate high-risk drivers [21]. Despite multiple tests evaluating cognitive decline and driving cessation [22], there is weak evidence in the current literature on the added value of these tests in screening procedures for predicting driving cessation for health reasons.

## Objectives

The primary objective of our study was to prospectively evaluate to what extent the TMT, the CDT, the MoCA test and the UFOV test can predict driving cessation in drivers aged 70 years or more in a Cox proportional hazard regression model. A secondary objective was to evaluate the predictive value of combining 3 or more positive tests in predicting driving cessation during a median 4-year follow-up.

## Methods

### Study design

We designed and conducted a prospective cohort study, the GARAge study, in collaboration with the State Driver and Vehicle Licencing Agency and the Swiss Automobile Club. All active drivers 70 years or older from predefined French-speaking regions of Switzerland were invited for a refresher course, and all drivers who participated in the course were invited to participate in the study to minimize bias. The median follow-up of these drivers was 4 years.

### Setting

The inclusion of participants took place between 2011 and 2013 in French-speaking regions of Switzerland (Lausanne, northern Vaud, Valais, Vevey, Montreux, Aigle and Entremont). Driving cessation was recorded by the official authorities, the State Driver and Vehicle Agency. Data were collected and registered in an independent way to eliminate bias through a median follow-up period of 4 years.

## Participants

All drivers aged 70 years of age or more who were residents of the seven defined regions of Switzerland (n = 16,858) were invited to a driving refresher course and to participate in the study. A total of 1,004 drivers agreed to participate in the refresher course. The participants were included in the study between May 2011 and September 2013 with a variable follow-up period ranging from 3 to 5 years and a median follow-up of 4 years. A total of 441 drivers fulfilled the inclusion criteria and agreed to participate. To participate, drivers had to be aged 70 years or more, have a valid Swiss driver's license, and not be institutionalized.

All participants were informed in detail about the procedures of the cohort and the data management. They were provided and signed a written informed consent form before inclusion in the study, in accordance with the ethical standards of the amended Declaration of Helsinki of 2008, Seoul. The participants voluntarily completed a driving test administered by the TSC (Auto Club Suisse) on the road during the day. Each participant completed a questionnaire with personal and health-related information, as well as information about driving habits. The study protocol was validated by the cantonal commission of human research ethics of the Vaud region, Switzerland (www.cer-vd.ch) and registered under the reference number CE 157/11.

## Exposure

We used basic and widely accepted scores (CDT, MoCA, TMT, UFOV) to assess the functional cognitive and visual capacities of older individuals and indirectly assess and predict their driving decline or cessation.

**Clock drawing test (CDT).** We calculated the CDT score using Freund's methods and the established 7-point scale [16]. Correct time indication is awarded up to 3 points, correct numbers are awarded up to 2 points and correct spacing is awarded up to 2 points. A positive CDS was defined as a score <5, a cutoff that showed significant positive and negative predictive values in previous studies [16]. Each score was evaluated twice, independently blinding the assessor to any other clinical information. Discrepancies were solved by an additional person.

**Montreal cognitive assessment (MoCA).** The MoCA was used to estimate the cognitive and functional status of the participants [23]. A cutoff of <26 was retained as a sensitive reference value for the diagnosis of mild cognitive impairment (MCI), following the current literature [15].

**Trail making test (TMT).** The TMT test consists of 2 parts. TMT-A calculates the time needed to connect 25 circles in ascending number order up to 25. In the TMT-B the participant has to connect 13 numbers and 12 letters alternatively in ascending numerical and alphabetical order [8]. Based on previous literature, we used the commonly accepted cutoffs of ≥54 sec for TMT-A or ≥150 sec for TMT-B [7].

**Useful field of view (UFOV).** The UFOV test is a computer-based visual and cognitive assessment and consists of 3 subtests. UFOV-1 tests processing speed, UFOV-2 tests processing speed during a divided attention task, and UFOV-3 tests processing speed for a selective attention task [11]. The reaction speed of the participants is measured in milliseconds (14–500 ms), and lower scores indicate better and faster performance on the tasks [24]. The participants were classified as low, moderate or high risk depending on their performance using the set standards from the UFOV. The dichotomization for survival analysis was accomplished by dividing the drivers into a moderate- to very-high-risk group and a low-risk group.

**Timed up and go (TUG).** We used a modified TUG test, as it is more adapted for clinical settings as a quick, reliable and easily performed functional evaluation tool for the elderly

population [25]. It has been shown to be one of the best performance tests for evaluating balance and functional impairment [26]. Mobility status is considered a strong predictor of TUG performance, and values below 12 seconds are used as the normal cutoff for elderly people 65 to 85 years old [27]. Cognitive impairment does not seem to affect the reliability of TUG test scores, and the variability of the results increases with the time needed to execute the test [28].

## Data sources/measurement

All participants were evaluated at baseline by independent staff, and information about medical status and driving events was recorded. Cognitive tests (TMT, CDT, MoCA test), ophthalmologic tests (UFOV) and mobility tests were performed for each participant. Data were collected on paper and in electronic form and double checked to minimize systematic errors. Data were collected and registered in an independent way to eliminate bias through a follow-up period of 4 years.

The on-road driving evaluation was performed by twelve certified driving instructors who participated in the study, either as self-employed or as employees of the Swiss Automobile Club. The routes were standardized for participants from the same region, were validated by the Swiss National Council for Road Security and were adapted the current Switzerland traffic control and examination standards. In order to minimize bias, all driving instructors were blinded to the psychometric and functional characteristics of the participants. Details about the methods used for the evaluation of driving performance can be found in our previous publications [7].

## Case definition

The outcome was defined as driving cessation caused by accident, voluntary cessation or cessation following medical advice. The outcome was recorded in an objective way, as it was provided by the official authorities and extracted from the State Driver and Vehicle Licencing Agency from the 12th of July 2016 to the 16th of August 2016 to minimize bias. Due to this method, the median follow-up of the study participants was 4 years. After the end of the trial, the dataset was locked and analyzed by an independent statistician. The statistical methods were predefined before the study and the multiplicity of analysis was taken into account. To reduce publication bias, all the results were analyzed and published.

## Sample size

Sample size was estimated using the log-rank test (Freedman method). Significant level was set at 0.05, and power was set at 0.8. The study was powered to detect an HR of 2, with an expected 90% of control subjects continuing to drive versus 80% for control. Expecting a ratio of 2 to 1 for control vs cases and a loss to follow-up of 5%, 483 participants should be recruited. In total, 441 drivers were included in the follow-up and the final analysis, which rendered our study sample underpowered for the expected results.

## Statistical methods

Hazard ratios were measured using Cox proportional hazards regression. We tested the assumption of proportionality using the Nelson Aalen estimate for the cumulative hazard function, concluding that the assumption was met. Patients who moved out of the canton, died or ceased driving for another reason were censored. We ran a secondary Cox regression model adjusting for age and sex. Furthermore, using Breslow methods, we verified whether

the incidence of events was constant over time. We transformed all continuous variables to range from 0 to 1 for the 25th and 75th percentiles of the studied population.

Following the current literature, we considered the MoCA, TMT, UFOV, CDT, and TUG as a battery of tests [29]. This would allow us to address the secondary objective of the study and evaluate the hypothesis of the predictive power of regrouping at least three positive tests regarded as a single test, as recommended by the literature [30]. Those with three or more positive tests were compared to those with two or fewer positive tests using an unadjusted and an adjusted Cox proportional hazard regression analysis [31].

The study protocol defined the statistical methods before the statistical analysis, which was performed by the STATA16 program. All data and statistical analysis files are available in Zenodo under the https://doi.org/10.5281/zenodo.4717755, following the current guidelines.

## Results

### Baseline characteristics/descriptive data

All 16,858 active drivers, aged 70 years or more, of French-speaking regions of Switzerland were invited to participate in a driving refresher course. From 1,004 course participants, 441 were included in the median 4-year follow-up (Fig 1).

The baseline characteristics of the sample were recorded at inclusion. Of the 441 participants, 216 (49.0%) were considered exempted from any health condition known to affect driving. Population baseline characteristics and health profiles are provided in **Table 1**.

Concerning the driving distance per week, the majority of the participants (53.3%) declared driving more than 200 km per week. The remaining 46.7% of the drivers declared a distance less than 150 km per week. A total of 73.2% of the participants had no accident during the previous 2 years, and 20.4% had 1 accident. The remaining 6.4% had 2 or more accidents on their

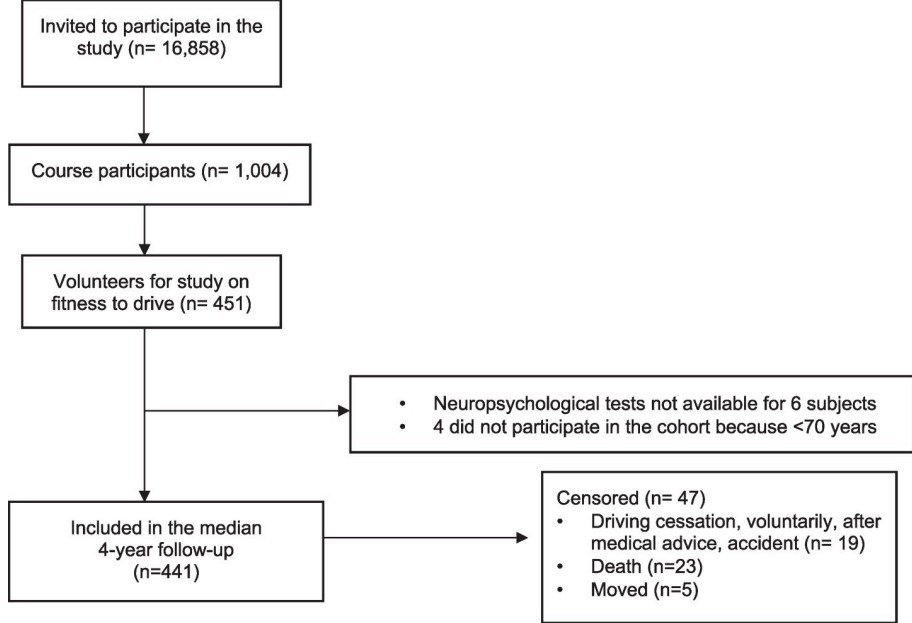

**Fig 1. Cohort flow chart of study participants.** From 1,004 course participants, 441 were included in the median 4-year follow-up. MoCA: MoCA: Montreal Cognitive Assessment, EPFL: École Polytechnique Fédérale Lausanne, n = sample size.

**Table 1. Baseline characteristics–population description.**

| Variables | All participants (n = 441) |
|---|---|
| | Mean (95% CI)/% (n) |
| Age (years) | 76 (75.5–76.4) |
| Sex (%male) | 61.2% (270) |
| Visual acuity (decimal) | 1.02 dec.(1–1.04) |
| Visual field (degreesº) | 182.9º (181.3–184.6) |
| Timed Up and Go (sec) | 8.5 sec (8.2–8.7) |
| MoCA$^{ca}$([0–30] points) | 26.4 (26.2–26.7) |
| MoCA$_{mod}$$^{B}$([0–29] points) | 23.6 (23.4–23.9) |
| **Driving distance per week** | |
| ≥200 km | 53.3% (234) |
| 150–199 km | 12.3% (54) |
| 100–149 km | 19.4% (85) |
| 50–99 km | 10.7% (47) |
| <50 km | 4.3% (19) |
| **Accidents previous 2 years** | |
| 0 | 73.2% (323) |
| 1 | 20.4% (90) |
| 2 | 4.3% (19) |
| 3 | 1.1% (5) |
| 4 | 0.23% (1) |
| 5 | 0.45% (2) |
| 8 | 0.23% (1) |
| **Driving performance** | |
| Excellent | 47% (197) |
| Good | 26.5% (111) |
| Moderate | 21.2% (89) |
| Poor | 5.3% (22) |
| **Clock drawing (Freund)** | |
| 1 | 0.23% (1) |
| 2 | 1.4% (6) |
| 3 | 5.4% (24) |
| 4 | 7.9% (35) |
| 5 | 22.9% (101) |
| 6 | 36.7% (162) |
| 7 | 25.4% (112) |
| **Clock drawing test** | |
| Positive (<5) | 15% (66) |
| Negative (≥5) | 85% (375) |
| **UFOV** | |
| Low risk | 70.8% (312) |
| Moderate risk | 20.9% (92) |
| High risk | 5.7% (23) |

Baseline characteristics of all drivers included in the cohort follow-up.

[a]MoCA = Montreal Cognitive Assessment.

[b]MoCA$_{mod}$ = Modified MoCA (without TMT or education level).

driving records. The on-road driving performance was estimated to be excellent for 47%, good for 26.5%, moderate for 21.2% and poor for 5.3% of the participants.

The CDT was found to be positive at baseline in 15% of the drivers (66 participants), with a positive test defined as a score less than 5. The mean MoCA score was 26.4 (95% CI: 26.2–26.7) at baseline, and the modified MoCA score was 23.6 (95% CI: 23.4–23.9). The TUG test was normal for all individuals at baseline (8.5 sec, 95% CI: 8.2–8.7).

## Survival

A total of 1738.5 person-years were observed within the study. The hazard regression survival analysis showed that the risk of driving cessation in a median follow-up of 4 years was 2.89 times higher for drivers who had a CDT<5 at baseline than for those with a score of CDT>5 (hazard ratio 2.89, 95% CI: 1.01–7.71, p = 0.033) (Fig 2A). Drivers who had a TMT-A $\geq$54 sec or TMT-B $\geq$150 sec at baseline had a 3 times higher risk of driving cessation than those with lower TMT scores (hazard ratio 3, 95% CI: 1.16–7.78, p = 0.023) (Fig 2B). The risk of driving

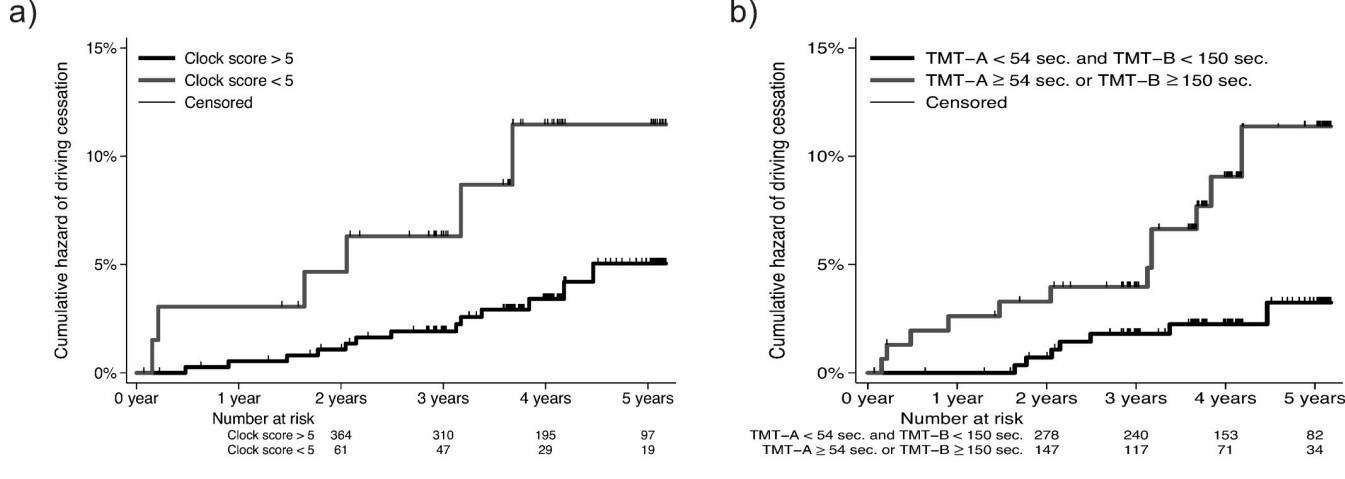

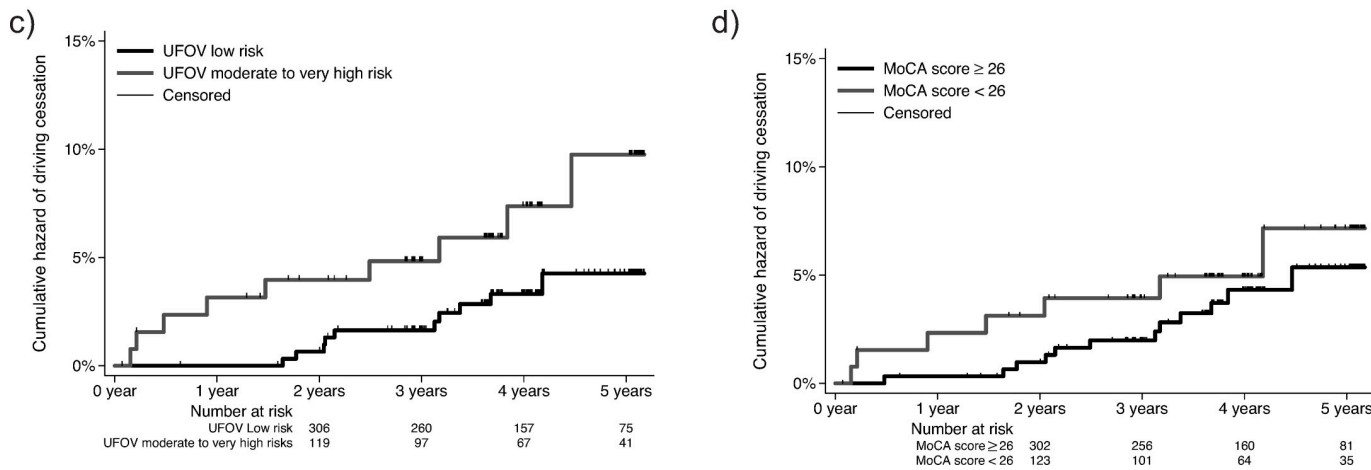

**Fig 2. Cox proportional hazards regression of driving cessation depending on baseline functional status.** Cox proportional hazards regression of driving cessation during a median 4-year follow-up depending on baseline: a) CDT score, b) TMT-A and TMT-B score, c) UFOV, and d) MoCA score. CDT: Clock Drawing Test, TMT: Trail Making Test, UFOV: Useful Field of View, MoCA: Montreal Cognitive Assessment.

**Table 2. Cox proportional hazards regression for driving cessation.**

| | Prevalence (n) | Unadjusted analysis[a] | | Adjusted analysis[b] | |
|---|---|---|---|---|---|
| | | HR (95%CI[c]) | P-value[d] | HR (95%CI[c]) | P-value[d] |
| • CDT < 5 | 15% (66) | 2.89 (1.1–7.62) | **0.031** | 2.89 (1.01–7.71) | **0.033** |
| • MoCA < 26 | 29.7% (131) | 1.41 (0.56–3.59) | 0.467 | 1.24 (0.48–3.18) | 0.660 |
| • TMT (A ≥54 sec. or B ≥150 sec.) | 35.4% (156) | 3.44 (1.35–8.76) | **0.009** | 3 (1.16–7.78) | **0.023** |
| • TUG (≥12 sec.) | 10.9% (48) | 1.95 (0.56–6.74) | 0.292 | 1.83 (0.52–6.4) | 0.343 |
| • UFOV moderate-to-high risk | 29.3% (129) | 2.29 (0.93–5.63) | 0.072 | 1.9 (0.75–4.82) | 0.178 |
| Battery at least 3 positive tests[e] | 14% (62) | 3.46 (1.31–9.13) | **0.012** | 3.12 (1.15–8.55) | **0.026** |

[a]Unadjusted Cox proportional hazard regression for driving cessation for CDT, MoCA, TMT, TUG and UFOV at a median 4-year follow-up.

[b]Adjusted Cox proportional hazard regression for CDT, MoCA, TMT, TUG and UFOV, adjusted for sex and age category, in a median 4-year follow-up

[c]CI: Confidence Interval.

[d]P-value of Cox regression–Breslow method for ties, evaluating the hazard ratio of drivers with low performance in the respective tests compared to high-performance drivers.

[e]Battery at least three positive tests out of five (MoCA, TMT, UFOV, CDT, TUG) considered as a single test.

CDT: Clock Drawing Test. MoCA: Montreal Cognitive Assessment. TMT: Trail Making Test. TUG: Timed Up and Go test. UFOV: Useful Field of View.

cessation in a median follow-up of 4 years was 1.9 times higher for drivers who had a moderate-to-high UFOV risk at baseline than for those with a low UFOV risk (hazard ratio 1.9 (95% CI: 0.75–4.82, p = 0.178) (Fig 2C) and 1.24 times higher for drivers who had an MoCA score < 26 at baseline than for those with a higher MoCA score (hazard ratio 1.24, 95% CI: 0.48–3.18, p = 0.660) (Fig 2D).

## Unadjusted analysis

The unadjusted hazard of driving cessation was found to be 2.89 times higher for drivers who with a CDT<5 at baseline than for those with a CDT>5 (hazard ratio 2.89, 95% CI: 1.1–7.62, p = 0.031), as shown in Table 2. The risk of driving cessation was estimated to be 3.44 times higher for drivers who had a TMT-A ≥54 sec or a TMT-B ≥150 sec at baseline than for those who had lower TMT scores (hazard ratio 3.44, 95% CI: 1.35–8.76, p = 0.009). The difference was statistically significant between drivers with low CDT and TMT score performance at baseline regarding the prospective risk of driving cessation.

The risk of driving cessation in a median follow-up of 4 years was calculated as 2.29 times higher for drivers who had a moderate-to-high UFOV risk at baseline than for those who had a low UFOV risk (hazard ratio 2.29, 95% CI: 0.93–5.63, p = 0.072). Furthermore, we found that the risk of driving cessation was 1.41 times higher for drivers who had an MoCA score < 26 at baseline than for those who had an MoCA score in the normal range (hazard ratio 1.41, 95% CI: 0.56–3.59, p = 0.467). The Cox proportional hazards regression of driving cessation did not show a statistically significant difference based on UFOV or MoCA scores at baseline.

We analyzed the predictive value of a battery of tests, as it would be impossible to test a model including multiple tests simultaneously, as only 4.3% of the drivers ceased driving for health reasons. Suspicious cases were defined arbitrarily by considering at least three positive tests out of five (MoCA, TMT, UFOV, CDT, TUG) regarded as a single test. We found that the cumulative hazard of driving cessation was significantly high in the unadjusted analysis for drivers who had a battery of at least three positive tests (unadjusted HR 3.46, 95% CI: 1.31–9.13, p = 0.012) (Fig 3).

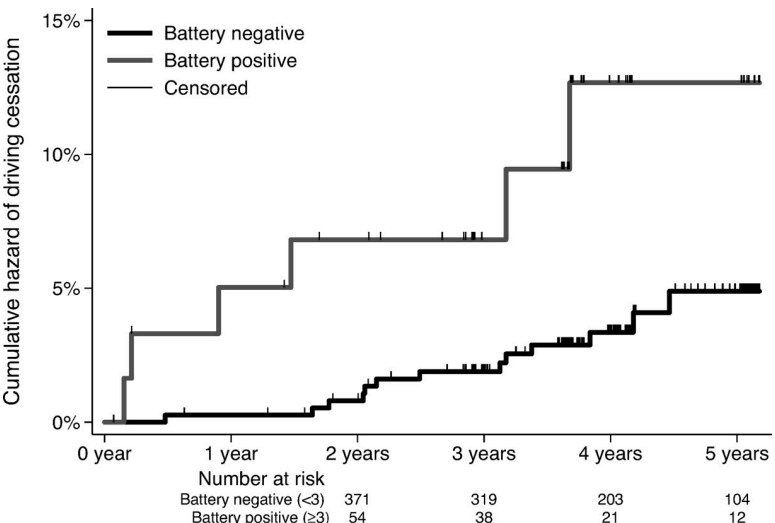

**Fig 3. Cox proportional hazards regression of driving cessation depending on baseline battery testing.** Cox proportional hazards regression of driving cessation during a median 4-year follow-up depending on a battery by considering at least three positive tests out of five (CDT, TMT, UFOV, MoCA, TUG) regarded as a single test. CDT: Clock Drawing Test, TMT: Trail Making Test, UFOV: Useful Field of View, MoCA: Montreal Cognitive Assessment. TUG: Time Up and Go test.

### Adjusted analysis

We adjusted the Cox proportional hazard regression analysis by taking into consideration age and sex (Table 2). The adjusted analysis suggests that the hazard ratio for driving cessation in older drivers of the same age, same sex and with a CDT score < 5, compared to those with a score ≥5, is 2.89 (95% confidence interval 1.01 to 7.71) during a median 4-year follow-up. This average 2.89-fold elevation in hazard for driving cessation is statistically significant, as the confidence interval, even though relatively large, does not include the value 1 and the p value is small (p = 0.033). Therefore, there is evidence of a difference in the hazard of the outcome between the two groups, in favor of the group performing better on the CDT, when adjusting for age and sex. The hazard ratio for driving cessation is 3 times higher for drivers with a TMT-A score ≥54 sec or a TMT-B ≥150 sec, when adjusting for age and sex, than for those with better TMT times. The 95% confidence interval ranges between 1.16 and 7.78, showing a statistically significant difference (p = 0.023). Although the risk of driving cessation seemed higher for drivers with an MoCA score <26 at baseline (HR 1.24, 95% CI: 0.48–3.18, p = 0.660), for drivers with a TUG test ≥12 sec (HR 1.83, 95% CI: 0.52–6.4, p = 0.394) and for drivers with a UFOV of moderate-to-high risk (HR 1.9, 95% CI: 0.75–4.82, p = 0.178), there was no evidence for a statistically significant difference.

Concerning the battery testing analysis, the cumulative hazard of driving cessation was significantly high in the adjusted analysis for drivers who had a battery of at least three positive tests (adjusted HR 3.12, 95% CI: 1.15–8.55, p = 0.026). The adjusted analysis is consistent with the results of the unadjusted analysis with similar hazard ratios and confidence intervals, and the observed associations were independent of sex and age.

### Discussion

The median 4-year follow-up of the study participants showed that the CDT and the TMT may predict driving cessation in a statistically significant way, with a better performance than the UFOV and MoCA tests. The participants with a TMT-A < 54 sec or a TMT-B < 150 sec

had a significantly lower cumulative hazard of driving cessation than those with a TMT-A $\geq$ 54 sec or a TMT-B $\geq$ 150 sc. The participants with a CDT score > 5 had a significantly lower cumulative hazard of driving cessation than those with a CDT score < 5. Similarly, an MoCA score $\geq$ 26 or a UFOV of low risk showed a lower cumulative risk at a median 4-year follow-up than an MoCA score <26 or a UFOV of moderate-to-high risk, although there was no evidence of a statistically significant difference, perhaps due to the limited power of the study.

## Strengths and limitations

One of the strengths of the current study was the representativeness of the sample, as we invited all active drivers aged 70 years or older who were registered by the official automobile authorities in a considerable region of Switzerland. We finally included in the follow-up those interested in the cognitive screening and the refresher course, elements that render our sample a representative sample of the elderly driver population in the primary care ambulatory setting of Switzerland. The inclusion procedure in the cohort minimizes any possible selection bias and contributes to the internal and external validity of the study.

Another strength of the current study was the prospective follow-up of a median of 4 years, which diminishes bias. The outcome of driving cessation was registered in an objective way by independent personnel in the setting of the official driving authorities, the State Driver and Vehicle Licencing Agency in Switzerland. The data were registered in an independent database with the information provided by the official authorities concerning driving cessation due to medical decisions or voluntary cessation. The independent and objective source of information concerning the outcome of the cohort renders the results more plausible, minimizing any potential source of bias and thus increasing the external validity of the study.

One of the major limitations of the study was that we did not adjust for confounding factors other than age and sex. Other residual confounding factors may exist, and the results should be interpreted with precaution. The outcome of interest, driving cessation, could have been based on medical decisions relying on tests similar to those defining exposure. Less subjective outcomes such as on-road accidents or deaths would have been more relevant but require including many more participants. The study was not powered to evaluate these outcomes. Furthermore, the current study does not provide any evidence of a causal relationship between the evaluated cognitive tests and driving cessation but shows a risk association depending on the baseline performance of basic primary care tests, such as the MoCA, the CDT, the UFOV and the TMT, in a prospective way. We can consider the results as an indicator of the strength of associations between initial tests and driving cessation during a median 4-year follow-up. The strength and significance level of the association is therefore by no means an indicator of the clinical importance of the tests. Indeed, for the studied tests, the number of false positives would be a better indicator of the avoidable societal burden from the procedure.

The fact that our study was underpowered consists of another major limitation of the results and statistical significance. The sample size estimated using the log-rank test (Freedman method) for a significance level of 0.05 and a power of 0.8 to detect an HR of 2 was 483 participants. We finally included 441 drivers in the follow-up, and the final analysis could have influenced the expected results. A low MoCA test and an UFOV of moderate-to-high risk seem to increase the risk of driving cessation, although they do not reach the expected level of statistical significance, perhaps due to the lack of power of the study. Furthermore, driving cessation events were rare, rendering the current prospective cohort study relatively underpowered.

We should mention that our results do not take into consideration the specific clinical situation of driving cessation and the reasons for medical decisions to stop driving for elderly

participants. We independently evaluated driving cessation either voluntarily or due to medical decisions, independent of the medical reasons that could have led to this decision. This lack of information was due to the absence of medical details that led to driving cessation by the official authorities' registries. Furthermore, our study was underpowered to detect any potential effect and relationship with medical reasons for driving cessation. It is important to note that health status can change over time and results from clinical tests might be affected. The study therefore does not rule out that participants exposure status might have changed over time.

Aside from the number of motor vehicle accidents, the number of citations (warnings or traffic tickets), the driving errors or traffic violations, could contain valuable information of fitness to drive. Current research shows that older drivers' driving errors may not be related to functional performance in healthy older adults [32]. One of the limitations of the current study is that we do not have information about the driving risks or the behavior of the participants. As this analysis was not included in the objectives of our study, the sample size was not powered enough in order to take into account the potential driving errors or violations and our results should be interpreted by taking into account this limitation. Furthermore, like most neuropsychological tests, the TUG is not only dependent of cognitive functions, as pain and reduced mobility can also affect results. This is the case for all tests, and neuropsychological tests need to be interpreted with care when assessing fitness to drive.

Another limitation of the current study, which is important for the evaluation of the external validity of the results, is the age limit of the study participants. The Cox proportional analysis and the results apply a sample of elderly drivers aged 70 years or more. We do not know whether the cognitive tests used in our study can play a predictive role or show an association with driving cessation prospectively in younger drivers. Thus, the results of this study cannot be applied to populations younger than 70 years old. Furthermore, our sample included patients from rural remote areas as well as cities, and this be considered when assessing the external validity limitations of the results, as driving cessation may be easier in rural areas. Public transports are highly developed in Switzerland even in rural areas making transportation available to most citizens.

Concerning battery testing, the combination of one or more tests can enhance the specificity of prediction, as shown by previous literature [30]. As most tests used in the current study are independent, our estimate of hazard and predictive values can be considered valid. However, a limitation of the current study is that the CDT is a part of the MoCA test and their combination in battery testing could influence the results. Even in this case of possible "convergence", as explained by Parikh et al. [30], the use of three tests would have minimal clinical significance but should be taken into consideration in the interpretation of the battery Cox proportional hazard analysis.

## Interpretation

Although our results are consistent with current evidence, they should be interpreted with precaution, taking into consideration the limitations of the current study. The results of the study show an association between low performance on the baseline tests of the MoCA, UFOV, CDT and TMT and a relatively higher risk of driving cessation prospectively. Recent studies show the importance of working memory and other cognitive functions, as patients with MCI have a higher prevalence of driving cessation [33]. Poor performance on diagnostic tests, such as the digit span backward test, has been shown to be associated with prospective driving cessation (OR: 0.493, 95% CI: 0.258–0.939) [33], although other tests, such as the MMSE, have not shown a significant effect [34].

Recent studies suggest that the MoCA could be a valuable tool for identifying fitness to drive, especially when adjusting for factors such as age, sex and walking speed [35]. These results are in accordance with our findings that show a 1.24-fold higher risk for driving cessation for elderly individuals who have an MoCA score < 26 at baseline than for those with a higher MoCA score, although we did not reach statistically significant results. A reason may be that our study was underpowered or a different cutoff should be considered, as suggested by sensitivity analysis showing an abnormal MoCA cutoff score < 28 [35].

Poor clinical evaluation by screening tools, such as the MMSE, TMT and CDT, seems to be independently associated with unsafe driving, defined as committing hazardous errors and/or traffic violations, and with restricted driving, defined as committing traffic or rule violations under certain driving conditions in a retrospective cross-sectional study [36]. Our prospective cohort study supports this hypothesis, as a driver with a CDT score <5 at baseline may have a 2.89 times higher risk of prospective driving cessation at 4 years than a driver with a CDT score >5 (p = 0.033).

Furthermore, recent studies show that drivers with a higher TMT test are associated with unsafe or restricted driving in a statistically significant way [36], while others argue that the TMT may not be specific enough in clinical practice to predict driving cessation without other investigations [7, 37]. Slow psychomotor speed and visual perception were associated with stopping or reducing driving in a one-year prospective cohort study with a significant OR of 1.15 (95% CI: 1.03–1.28) [38]. Our findings are in line with the current evidence, showing that a driver with a TMT-A ≥54 sec or a TMT-B ≥150 sec at baseline may have a threefold risk of driving cessation compared to those with lower TMT scores (p = 0.023). However, there is lack of evidence supporting the use of neuropsychological tests for screening purposes in assessing fitness to drive. Our study suggests most tests to have low predictive value for correctly classifying patients in risk levels for future driving cessation. Adapted ecological on-road tests might be more valuable in correctly assuming patients are unfit to drive, taking into account the screening limitations and methods to overcome them [39].

Additionally, evidence shows that instrumental cognitive and functional status is important when assessing fitness to drive, in association with visual and constructional functioning and visuospatial abilities [34]. A ten-year follow-up prospective cohort study with 1248 participants showed that slower speed of processing as measured by the UFOV test could be a predictor of driving cessation (HR 1.76, p<0.01) [40]. Our findings seem to be in accordance with previous evidence, suggesting that a low-risk UFOV may be associated with a lower risk of driving cessation, although the results did not reach statistical significance (HR 1.9, p = 0.178).

The current literature on the evaluation of a battery of neurocognitive tests for the prediction of driving cessation is limited. Our findings that the combination of at least three out of five positive tests (MoCA, TMT, UFOV, CDT, TUG) may be associated with an increased hazard of driving cessation in the adjusted and unadjusted analysis (p<0.05) could be a useful tool for risk assessment in the primary care setting, taking into consideration the limitations mentioned above.

According to current evidence, neuropsychological and visuospatial tests can play a major role in screening elderly drivers 70 years old or older, but they should not be considered discriminating or decisive for their actual or future driving capacity [41]. They can show an association between low cognitive test performance and driving cessation prospectively, but this association is not direct and not causal. Thus, we can conclude that no test seems to independently predict driving cessation with reliable predictive values, and more specific tests and on-road driving performance evaluations should be taken into consideration by the official authorities when determining driving authorization in the elderly population.

## Practical implications

The results of our study can be generalized to the elderly Swiss population by taking into consideration the baseline characteristics of the participants of the study. The participants were drivers 70 years old or older with an active driving license from French-speaking Switzerland areas. The sample can be considered representative of the general population, as we invited all the individuals who had active driving licenses following the official authority records, minimizing the risk of selection bias. Thus, our sample can be generalized to drivers requiring systematic screening in the primary care setting in ambulatory health care, where the abovementioned tests are accessible and can easily be performed.

Although tests such as the TMT, UFOV, MoCA and CDT may be associated with future driving cessation, individually or in batteries of at least three positive tests, we do not have any evidence that these tests can independently predict driving cessation with reliable predictive values [42]. There is a lack of evidence on the validity of neuropsychological tests to correctly identify fitness to drive, therefore they only provide indicators of associations without giving indications of validity [43]. Further research is needed to define or develop specific tests in association with on-road driving performance evaluations that could be taken into consideration by the official authorities when determining driving authorisation in the elderly population.

## Conclusion

Our study confirms previous findings that neuropsychological tests may be associated with driving cessation. However, predicting driving cessation by cognitive and visual tools in the primary care setting remains a challenge. The TMT, the CDT, the MoCA test and the UFOV could be considered possible tools to estimate the risk of driving cessation in drivers 70 years old or older. Although the CDT and the TMT seem to predict driving cessation in a statistically significant way, with a better performance than the UFOV and MoCA tests, the association remains weak for a 4-year period, and further research is required before setting cutoff values to impose driving cessation based on these tests alone. A battery of at least three positive tests may also be associated with a higher hazard of driving cessation. Although our results are consistent with current evidence, they should be interpreted with precaution, taking into consideration the limitations of the study. Further research is needed to define specific screening tests in association with on-road evaluations as possible and reliable predictors of driving cessation in elderly patients.

## Acknowledgments

We gratefully thank Frederique Margot for organizing appointments with the participants, Sylvie Hautle and the Swiss Automobile Club for the refresher course organization and facility concession.

## Author Contributions

**Conceptualization:** Paul Vaucher, Bernard Favrat.

**Data curation:** Paul Vaucher, Isabel Cardoso.

**Formal analysis:** Ioannis Kokkinakis, Paul Vaucher, Bernard Favrat.

**Funding acquisition:** Paul Vaucher, Bernard Favrat.

**Investigation:** Paul Vaucher, Isabel Cardoso, Bernard Favrat.

**Methodology:** Ioannis Kokkinakis, Paul Vaucher, Isabel Cardoso, Bernard Favrat.

**Project administration:** Paul Vaucher, Bernard Favrat.

**Resources:** Paul Vaucher, Bernard Favrat.

**Software:** Ioannis Kokkinakis, Paul Vaucher.

**Supervision:** Paul Vaucher, Bernard Favrat.

**Validation:** Ioannis Kokkinakis, Paul Vaucher, Bernard Favrat.

**Visualization:** Ioannis Kokkinakis, Paul Vaucher.

**Writing – original draft:** Ioannis Kokkinakis.

**Writing – review & editing:** Ioannis Kokkinakis, Paul Vaucher, Isabel Cardoso, Bernard Favrat.

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
