## [Decision Letter · Decision Letter 0]

2 Jun 2021

PONE-D-21-15340

Assessment of cognitive screening tests as predictors of driving cessation: a prospective cohort study of a median 4-year follow-up

PLOS ONE

Dear Dr. Kokkinakis,

Thank you for submitting your manuscript to PLOS ONE. After careful consideration, we feel that it has merit but does not fully meet PLOS ONE’s publication criteria as it currently stands. Therefore, we invite you to submit a revised version of the manuscript that addresses the points raised during the review process.

We look forward to receiving your revised manuscript.

Kind regards,

Abiodun E. Akinwuntan, PhD, MPH, MBA

Academic Editor

PLOS ONE

Journal Requirements:

Reviewers' comments:

Reviewer's Responses to Questions

**Comments to the Author**

1. Is the manuscript technically sound, and do the data support the conclusions?

Reviewer #1: Yes

Reviewer #2: Yes

2. Has the statistical analysis been performed appropriately and rigorously? 

Reviewer #1: Yes

Reviewer #2: Yes

3. Have the authors made all data underlying the findings in their manuscript fully available?

Reviewer #1: Yes

Reviewer #2: Yes

4. Is the manuscript presented in an intelligible fashion and written in standard English?

Reviewer #1: Yes

Reviewer #2: Yes

5. Review Comments to the Author

Reviewer #1: Thank you for offering an opportunity to review an article titled “Assessment of cognitive screening tests as predictors of driving cessation: a prospective cohort study of a median 4-year follow-up”. This article investigated the driving cessation predictability based on various clinical tests in drivers aged 70 or higher. Overall, the manuscript is well written. However, there are still some concerns as stated below. Therefore, I would like to recommend a major revision for the current manuscript.

Abstract

Well written.

Introduction

1. Background information of clinical tests were well described.

2. Lack of discussion on driving cessation based on previous literature. For example, readers may also want to know when driving cessation usually happens, what the relationship between driving cessation and individual health looks like, what would the impact of driving cessation looks like from an individual perspective as well as societal perspective, etc. If authors can include an in-depth discussion (perhaps, create one new paragraph), it would enhance the current manuscript and help readers catch the importance of the manuscript more easily.

3. Study objectives are clearly presented.

Methods

1. All the clinical tests utilized in the study were clearly demonstrated.

2. Other methodologies are well described.

Results

1. It seems like driving performance (Table 1) was assessed by driving evaluation officers (?). Until Table 1 in the Results, I was not aware of how driving performance was assessed. If there are any relevant information how it was determined, the authors may want to add a short description in the methods.

2. Aside from the number of motor vehicle accidents, sometimes the number of citations (warnings, or traffic tickets) could contain valuable information of fitness to drive. If there was any, try to add them in the results. If not, the authors may want to add this point in the limitation section.

3. In the Table 1, participant classification depending on the three outcome categories of UFOV outcomes (low, moderate, high) are not presented. Was there any reason? Otherwise, this information will help readers understand the demographics.

4. The authors divided the whole group into two groups, healthy and unhealthy. What made the authors classify those two groups? While I was reading the paragraph (Lines 238 to 248). I was expecting to see any analysis (i.e., a survival analysis depending on healthy vs unhealthy status), but I do not think there is any in the current manuscript.

Discussion

Good summary of findings and good discussion of strengths, limitations, interpretation, and practical implications.

Conclusion

Well-written

Minor comments

Line 52: need a space between 1.16 and -.

Line 192: What is the class III?

Line 472: MoCA

Reviewer #2: Summary:

This study is a prospective cohort study of older drivers aged 70 and older from Switzerland who volunteered to partake in an education session in relation to driving, then complete an on road driving evaluation and associated neuropsychology based testing. The participants were then followed through the motor licensing Bureau to establish whether driving cessation had occurred for the reasons of either cessation due to health reasons, cessation due to physician recommendation, or health related collision. The mean time of follow-up for participants was 4 years. Hazard ratios were used to identify that certain predictors including Trail making test part A and B as well as the clock drawing test were correlated with higher likelihood of driving cessation. A combination of the neuropsychology tests also identified drivers who were more likely to cease driving.

Major Comments:

1. Relevance or implications of the study results: This study has a very significant number of participants with the important outcome of driving cessation. However, one of the limitations of this study is the ability to interpret the results and apply them meaningfully at the individual level. Overall the authors describe only 4.3% of participants stopping driving due to medical reasons. The time for follow-up is at a mean time of 4 years from time of baseline assessment. Given the age of the population and the prolonged time course, it is not clear that the health status of these drivers would be stable enough such that their evaluation up to 4 years earlier would have contributed information to them ceasing driving- for instance new health events may have contributed to the cessation. The authors would need to make a clear how this information advances the field for assisting clinicians or older drivers themselves in making decisions in relation to driving cessation. For instance, whether "healthy" or “unhealthy” the idea of stopping driving based on these measures does not appear very effective for influencing decision-making. There clearly is a correlation with lower values on the neurocognitive tests however providing direction to the clinician or patient based on these tests seems unclear.

Review:

Title: Overall the title is clear and describes the study well.

Abstract: The abstract is accurate. It does reflect the content within the article and is consistent with the information. It is coherent, readable and structured.

Introduction: The background information is relevant to the study at hand. There is some commentary regarding tests (e.g. TUG test) that are not specifically neurocognitive in nature such as the timed up and go test which deviates from the overall aim of this manuscript/paper. The authors in the background do comment on sensitivity and specificity rates of tools in relation to driving prediction–this manuscript will not delve into the sensitivity or specificity or predictive ability of these tests specifically. Further the background does not touch upon the idea of the duration for which screening or assessment measures are valid. This is relevant and that the study will look at 4 years for the median time of follow-up.

Methods: The study design is a prospective cohort study where participants volunteered. There was no further testing after initial assessment at 1 year and a data collection in the form of driving cessation results was collected at the meantime approximately 4 years post onset of study. The study population and recruitment were well described. The statistical analyses were appropriate. One comment can be made upon the definition of "healthy and unhealthy" where it is noted that through this volunteer study that the authors classify the majority of participants as "unhealthy".

Results: The results of the study are clearly described. There is some overlap of the text and the tables but not to a significant degree. The tables are appropriate. The figures are quite blurry and difficult to read and interpret and overall have poor quality.

Discussion: Discussion overall is appropriate for the aims and results of the study. The authors do comment on pertinent literature. The main limitation of the discussion section is in relation to the value and utility of these results. While the authors indicate there is a correlation of these tests with eventual cessation of driving, they do not elaborate as to how this might further management of patients from either the clinician or patient perspective.

In relation to limitations of the study while the authors indicate that there is good representation of the population, other factors such as general health conditions or even if the drivers were rural versus urban living with a higher need for the ability to drive versus use of public transportation systems available to urban dwellers.

Conclusions: These do reflect to the findings of the study. They describe an association of the neuropsychology test with driving cessation.

6. PLOS authors have the option to publish the peer review history of their article (what does this mean?). If published, this will include your full peer review and any attached files.

Reviewer #1: **Yes: **Sanghee Moon

Reviewer #2: No

---

## [Author Response · Author response to Decision Letter 0]

28 Jul 2021

Modifications following the Reviewer Comments: 

Reviewer #1 ( Sanghee Moon ): 

1) “Introduction - Lack of discussion on driving cessation based on previous literature. For example, readers may also want to know when driving cessation usually happens, what the relationship between driving cessation and individual health looks like, what would the impact of driving cessation looks like from an individual perspective as well as societal perspective, etc. If authors can include an in-depth discussion (perhaps, create one new paragraph), it would enhance the current manuscript and help readers catch the importance of the manuscript more easily.”

Thank you for this very constructive and useful comment. We took into consideration your proposal and adapted the text by adding a paragraph (Lines 70 – 81). We highlighted that research has shown that maintaining out-of-home mobility is of great importance for people moving to old age from late midlife, as well as the association between driving cessation and functional dependency, depressive disorders, social dysfunction and mortality, with a considerable individual and societal impact. As many adverse health problems have been related to driving cessation in later life, predicting and evaluating accurately the decline of driving capacity of older drivers is of critical importance. Supporting and stimulating out-of-home mobility in the elder population, detecting and preventing a functional decline and possible future driving cessation, depends on individual screening strategies, as well as on transport policy and social policy measures. 

Indeed, including this discussion, it would enhance the current manuscript and help readers catch the importance of the research subject more easily.

2) “Results - It seems like driving performance (Table 1) was assessed by driving evaluation officers (?). Until Table 1 in the Results, I was not aware of how driving performance was assessed. If there are any relevant information how it was determined, the authors may want to add a short description in the methods.”

As described in our previous publications (Vaucher et al. 2014, DOI: https://doi.org/10.1186/1471-2318-14-123), we specified how driving performance was assessed and we added a short description in the methods, as proposed (Lines 203-210). The on-road driving evaluation was performed by twelve certified driving instructors who participated in the study, either as self-employed or as employees of the Swiss Automobile Club. The routes were standardized for participants from the same region, were validated by the Swiss National Council for Road Security and were adapted the current Switzerland traffic control and examination standards. In order to minimize bias, all driving instructors were blinded to the psychometric and functional characteristics of the participants. Details about the methods used for the evaluation of driving performance can be found in our previous publications.

3) “Results - Aside from the number of motor vehicle accidents, sometimes the number of citations (warnings, or traffic tickets) could contain valuable information of fitness to drive. If there was any, try to add them in the results. If not, the authors may want to add this point in the limitation section.” 

Thank you for this very important comment. Indeed, aside from the number of motor vehicle accidents, the number of citations (warnings or traffic tickets), the driving errors or traffic violations, could contain valuable information of fitness to drive. However, one of the limitations of the current study is that we do not have information about the driving risks or the behavior of the participants. As this analysis was not in the objectives of our study, the sample size was not powered enough in order to take into account the potential driving errors or violations. We added a related paragraph in the limitations section, as our results should be interpreted by taking into account this limitation (Lines 407 - 414). 

4) “Results - In the Table 1, participant classification depending on the three outcome categories of UFOV outcomes (low, moderate, high) are not presented. Was there any reason? Otherwise, this information will help readers understand the demographics”

Indeed, this information is very important for the internal validity, the consistency and the external validity of the study. We added the participant baseline UFOV classification in the Table 1. This information will effectively contribute to a better understanding of the demographics. 

5) “Results - The authors divided the whole group into two groups, healthy and unhealthy. What made the authors classify those two groups? While I was reading the paragraph (Lines 238 to 248). I was expecting to see any analysis (i.e., a survival analysis depending on healthy vs unhealthy status), but I do not think there is any in the current manuscript.”

Thank you for this important comment contributing to enhance the internal validity of the study. Separation of healthy and non-healthy population was done to test psychometric properties of the clock drawing task. However, this analysis is irrelevant for this publication and all referencing to the separation of groups has been removed. 

6) “Line 52: need a space between 1.16 and -.”

The space has been added as indicated. 

7) “Line 192: What is the class III?”

This section has been erased, as mentioned in the answer to the 5th question. 

8) “Line 472: MoCA”

The word was corrected as indicated. 

Reviewer #2: 

1) “Major Comment: Relevance or implications of the study results: This study has a very significant number of participants with the important outcome of driving cessation. However, one of the limitations of this study is the ability to interpret the results and apply them meaningfully at the individual level. Overall the authors describe only 4.3% of participants stopping driving due to medical reasons. The time for follow-up is at a mean time of 4 years from time of baseline assessment. Given the age of the population and the prolonged time course, it is not clear that the health status of these drivers would be stable enough such that their evaluation up to 4 years earlier would have contributed information to them ceasing driving- for instance new health events may have contributed to the cessation. The authors would need to make a clear how this information advances the field for assisting clinicians or older drivers themselves in making decisions in relation to driving cessation. For instance, whether "healthy" or “unhealthy” the idea of stopping driving based on these measures does not appear very effective for influencing decision-making. There clearly is a correlation with lower values on the neurocognitive tests however providing direction to the clinician or patient based on these tests seems unclear.”

Thank you very much for these comments and proposals that helped us enhance the consistency of the manuscript. All referencing to healthy or unhealthy drivers have been removed. This notion was used to test psychometric properties of the clock drawing test at baseline only. For the other analysis, the aim is to test the predictive properties of specific tests in correctly forecasting driving cessation. As such, other events affecting health might occur and change exposure status. This is however also the case in real life situations where fitness to drive is evaluated only every two years. The reviewer’s comment nevertheless is very relevant, and we have added a small statement in the limitation section.

Changes: “Health status can change over time and results from clinical tests might be affected. The study therefore does not rule out that participants exposure status might have changed over time” (Lines 403 – 406).

2) “Introduction: There is some commentary regarding tests (e.g. TUG test) that are not specifically neurocognitive in nature such as the timed up and go test which deviates from the overall aim of this manuscript/paper.”

The TUG test is considered as a clinical test for executive functions. Like most neuropsychological tests, the TUG is not only dependent of cognitive functions, as pain and reduced mobility can also affect results. This is the case for all tests, and neuropsychological tests need to be interpreted with care when assessing fitness to drive. This limitation was added in the methods section (Lines 414 – 416).

3) “Introduction: The authors in the background do comment on sensitivity and specificity rates of tools in relation to driving prediction. This manuscript will not delve into the sensitivity or specificity or predictive ability of these tests specifically.”

Sensitivity and specificity require having a gold standard to rely on. Driving cessation is not a gold standard for unfitness to drive. Systematic reviews on neuropsychological tests have shown low validity (Mathias et al. 2009, Bennett et al. 2016). However, these studies rely on outcomes that cannot be considered as gold standard of fitness to drive (e.i. on-road driving test, simulator driving test, driver problems). In our study, some people stop driving for other reasons or feel unfit without being unfit. We therefore only provide indicators of associations without giving indications of validity. This lack of evidence on the validity of neuropsychological tests to correctly identify fitness to drive was added to the discussion in the “practical application” section (Lines 500 - 507).

4) “Introduction: Further the background does not touch upon the idea of the duration for which screening, or assessment measures are valid. This is relevant and that the study will look at 4 years for the median time of follow-up.”

This comment is very relevant as it points out that exposure status could change over time. The manuscript has a pragmatic approach given that most countries screen drivers for fitness to drive at a given frequency (in Switzerland, once every two years) and that health events can also occur between visits. This study aims to test the predictive ability of these screening procedures over what actually happens in the following years. The limitation of the study in correctly classifying participants in the survival analysis has been added. The practical implications of this limitation has been added to the discussion. We focused on the difficulties of correctly assessing the state of health and its effects on driving over time. Screening constitutes a high risk of falsely preventing people from driving. Driving cessation is therefore conditional, and the health status affecting driving needs to be documented (screening leading to further investigations on cause, on-road evaluation).

5) “Methods: One comment can be made upon the definition of "healthy and unhealthy" where it is noted that through this volunteer study that the authors classify the majority of participants as "unhealthy".”

Notions of healthy and unhealthy have been removed (see answer to comment 1).

6) “Results: The figures are quite blurry and difficult to read and interpret and overall have poor quality.”

Thank you for your useful observation that allowed us to optimize the quality of the figures. Their quality seemed lower in TIFF format after using the PACE tool during our first submission. 

Changes: all figures have been provided in vectorized format using Adobe Illustrator as recommended by the Plos One instructions to the authors. They were adapted and exported to EPS format per Plos One requirements in order to achieve high publication quality figure presentation. Furthermore, all figures are also immediately available in TIFF format, adapted by PACE tool. 

7) “Discussion: Discussion overall is appropriate for the aims and results of the study. The authors do comment on pertinent literature. The main limitation of the discussion section is in relation to the value and utility of these results. While the authors indicate there is a correlation of these tests with eventual cessation of driving, they do not elaborate as to how this might further management of patients from either the clinician or patient perspective.”

A small paragraph was added on the lack of evidence supporting the use of neuropsychological tests for screening purposes in assessing fitness to drive. Our study suggests most tests to have low predictive value for correctly classifying patients in risk levels for future driving cessation. Adapted ecological on-road tests might be more valuable in correctly assuming patients are unfit to drive. Furthermore, it seems important to discuss how screening has limitations and how to overcome them. This was added to the discussion (Lines 465 – 470).

8) “Discussion: In relation to limitations of the study while the authors indicate that there is good representation of the population, other factors such as general health conditions or even if the drivers were rural versus urban living with a higher need for the ability to drive versus use of public transportation systems available to urban dwellers.”

This point is relevant and is indeed worth mentioning. It is less an internal validity issue rather than an external validity. Our sample included patients from rural remote areas as well as cities. Driving cessation is easier in rural areas in Switzerland. Compared to other countries such as Canada or Australia, driving cessation is less likely to isolate people. Public transports are highly developed in Switzerland making transportation available to most citizens. This was added to the discussion in the limitation section (Lines 422 – 426).

---

## [Decision Letter · Decision Letter 1]

9 Aug 2021

Assessment of cognitive screening tests as predictors of driving cessation: a prospective cohort study of a median 4-year follow-up

PONE-D-21-15340R1

Dear Dr. Kokkinakis,

We’re pleased to inform you that your manuscript has been judged scientifically suitable for publication and will be formally accepted for publication once it meets all outstanding technical requirements.

Kind regards,

Abiodun E. Akinwuntan, PhD, MPH, MBA

Academic Editor

PLOS ONE

Additional Editor Comments (optional):

Reviewers' comments:

Reviewer's Responses to Questions

**Comments to the Author**

1. If the authors have adequately addressed your comments raised in a previous round of review and you feel that this manuscript is now acceptable for publication, you may indicate that here to bypass the “Comments to the Author” section, enter your conflict of interest statement in the “Confidential to Editor” section, and submit your "Accept" recommendation.

Reviewer #1: All comments have been addressed

2. Is the manuscript technically sound, and do the data support the conclusions?

Reviewer #1: Yes

3. Has the statistical analysis been performed appropriately and rigorously? 

Reviewer #1: Yes

4. Have the authors made all data underlying the findings in their manuscript fully available?

Reviewer #1: Yes

5. Is the manuscript presented in an intelligible fashion and written in standard English?

Reviewer #1: Yes

6. Review Comments to the Author

Reviewer #1: The authors properly addressed reviewer's concerns and suggestions. I recommend to accept this manuscript in its present form. Thank you.

7. PLOS authors have the option to publish the peer review history of their article (what does this mean?). If published, this will include your full peer review and any attached files.

Reviewer #1: **Yes: **Sanghee Moon

---

## [Editor Report · Acceptance letter]

12 Aug 2021

PONE-D-21-15340R1 

Assessment of cognitive screening tests as predictors of driving cessation: a prospective cohort study of a median 4-year follow-up 

Dear Dr. Kokkinakis:

I'm pleased to inform you that your manuscript has been deemed suitable for publication in PLOS ONE. Congratulations! Your manuscript is now with our production department. 

Kind regards, 

on behalf of

Dr. Abiodun E. Akinwuntan 

Academic Editor

PLOS ONE